# Multi-Stage Attentive Network for Motion Deblurring via Binary Cross-Entropy Loss

**DOI:** 10.3390/e24101414

**Published:** 2022-10-03

**Authors:** Cai Guo, Xinan Chen, Yanhua Chen, Chuying Yu

**Affiliations:** 1Network and Educational Technology Center, Hanshan Normal University, Chaozhou 521041, China; 2School of Computing and Information Engineering, Hanshan Normal University, Chaozhou 521041, China; 3School of Physics and Electronic Engineering, Hanshan Normal University, Chaozhou 521041, China

**Keywords:** motion deblurring, multi-stage attentive network, binary cross-entropy loss

## Abstract

In this paper, we present the multi-stage attentive network (MSAN), an efficient and good generalization performance convolutional neural network (CNN) architecture for motion deblurring. We build a multi-stage encoder–decoder network with self-attention and use the binary cross-entropy loss to train our model. In MSAN, there are two core designs. First, we introduce a new attention-based end-to-end method on top of multi-stage networks, which applies group convolution to the self-attention module, effectively reducing the computing cost and improving the model’s adaptability to different blurred images. Secondly, we propose using binary cross-entropy loss instead of pixel loss to optimize our model to minimize the over-smoothing impact of pixel loss while maintaining a good deblurring effect. We conduct extensive experiments on several deblurring datasets to evaluate the performance of our solution for deblurring. Our MSAN achieves superior performance while also generalizing and compares well with state-of-the-art methods.

## 1. Introduction

Motion blur removal is a challenging problem in vision tasks due to the fact that motion blurs are spatially varying blurs caused by various factors, e.g., camera shaking or objects quickly moving [1]. The single-image deblurring issue is to restore the latent sharp image by the given blurred image. Traditional image-deblurring methods based on simplistic assumptions (e.g., blur kernel *K* is uniform, partially uniform, or locally linear) directly model the degradation process of image blurring as a convolution form:(1)$$I_{B}=k*I_{S}+b,$$
where $I_{B}$, $I_{S}$ and *b* are the blurred image, original sharp image, and noise, respectively. *k* is the blur kernel. The symbol * represents the operation of convolution.

Because the blur kernel in the blind image deblurring is unknown, blind deblurring methods need to estimate blur kernel *k* to restore latent sharp image $I_{S}$. Therefore, accurate prediction of the blur kernel is essential. Most traditional image-deblurring methods estimate the blur kernel by making full use of various prior information from natural images, e.g., non-locally prior [2], framelet based prior [3], $\ell^{0}$-regularization based prior [4,5], dark-channel based prior [6], and salient edges prior [7], etc.

However, most of these methods require expensive computations to estimate the unknown blur kernel due to complex heuristic parameter optimization algorithms. In particular, the most essential reason for motion blur is that the large-distance relative motion of objects in the shooting scene causes the originally collected pixels to change during the exposure time, and the variation of these pixels is complex and non-uniform [8,9,10]. These previous methods cannot deal with complex motion blurs well because finding different blur kernels for different pixels is a severely ill-posed problem.

Recent learning-based end-to-end deblurring methods show their strengths in addressing these issues because they are free of blur kernels. These end-to-end methods do not need to estimate blur kernels, rather only the mapping learned between blurred images and sharp images, so as to directly generate predicted restored images from the blurred images. In particular, Nah et al. [10] proposed a multi-scale deblurring neural network based on multi-stage networks (MSNs) for directly estimating latent sharp images from the blurred images, and other methods [11,12,13,14,15,16,17,18,19,20,21,22,23,24] further advance MSN models for deblurring. Moreover, generative adversarial network (GAN)-based methods [25,26,27] have also been used to achieve end-to-end image deblurring.

Although existing end-to-end approaches have made great progress in motion deblurring, there still exist two main issues. First, because the models learn the mapping of blurred and sharp image pairs is all single determination, most of these methods suffer from insufficient generalization ability (experimental results to be given in Section 4.2). Secondly, most MSN methods mainly use pixel loss to train their models. However, pixel loss such as mean square error (MSE) encourages models to find a reasonable pixel average-wise for the solution [28]. Models optimized with this loss function tend to produce over-smoothed outputs. Moreover, although GAN-based methods employ adversarial losses instead of pixel loss, they often require multiple losses and careful tuning of parameters [29].

To this end, we propose a multi-stage attentive network (MSAN) for motion deblurring via binary cross-entropy loss. We utilize attention mechanisms to improve the adaptability of our MSAN for different blurred images. We design a self-attention module (SAM) by using group convolution to be more suitable for our model to enhance the extraction of relevant features and suppression of irrelevant features in the deblurring process. Our model can recalibrate the features from different blurred images through SAM, thus improving the adaptability of the model to different blurred images. In this way, our method can adapt to different blurred scenes. Moreover, we artfully use binary cross-entropy loss instead of pixel loss to optimize our model to reduce over-smooth output results. In particular, the proposed MSAN is cascaded by four sub-networks and trained in an end-to-end manner without stage-wise optimization.

Notably, in comparison with our previous approach [23], this work has the following main differences. (1) We insert a self-attention module with group convolution between the encoder and decoder based on the sub-network structure of the method [23]. That is mainly inspired by [30,31], enhancing the adaptability of the model by computing the correlation of relevant positions to obtain the most useful and important features and recalibrate these features. (2) To reduce over-smooth output results, we introduce binary cross-entropy loss to train the model to deblur. (3) To demonstrate the validity of our proposed method, we conduct more comparative experiments by using synthetic and real-world datasets.

Our major contributions can be summed up as follows.

We propose a new attention-based end-to-end method on top of multi-stage networks. A self-attention module is introduced to improve the adaptability of the model to different blurred images. In particular, we apply group convolution to the self-attention module, which effectively reduces the computing cost of self-attention.We propose a novel strategy based on binary cross-entropy loss instead of pixel loss to optimize our model and to reduce the over-smooth impact by pixel loss while maintaining a good deblurring effect.We perform experiments on multiple representative datasets and provide extensive analyses and evaluations on the results. The results of experiments demonstrate that our MSAN outperforms SOTA methods when processing different types of blurred images.

## 2. Related Work

The purpose of this section is to describe learning-based end-to-end deblurring methods and attention mechanisms in image deblurring.

### 2.1. Learning-Based End-to-End Deblurring Methods

The learning-based deblurring methods avoid blur kernel estimation. These methods are trained with original sharp and artificially synthesized blurred image pairs for learning how to map blurred images and sharp images, so as to generate predicted restored images from the blurred images directly.

Nah et al. [10] proposed a multi-scale deblurring neural network and adopt ResBlock (a slightly modified version of the residual network structure) as the building block of the model. This model utilized three cascaded sub-networks to process images of different scales to simulate the optimization process from coarse to fine to gradually recover potentially sharp images. On the basis of this multi-scale method, Tao et al. [11] presented a network using scale-recurrent architecture with recurrent modules to reduce the model size and improve the training stability by sharing weight across different sub-networks. A later study by Gao et al. [12] suggested that a parameter-selective sharing scheme in combination with a nested skipping structure of the nonlinear transformation module further optimizes the multi-scale scheme and achieves more effective deblurring effects than [10,11]. Different from multi-scale schemes, Zhang et al. [13] proposed a hierarchical multi-patch network that takes advantage of deblurring cues at several scale patches. This multi-patch localized-to-coarse approach is used to perform deblurring in the fine-to-coarse grids. To further improve the performance of the multi-patch model, they also proposed a stacked version. A recent proposal by Cai et al. [18] has been to integrate dark and bright channel priors into the multi-stage network in order to achieve effective motion deblurring. By using the cascade architecture for single-image deblurring, Lim et al. [19] utilize the complementary characteristics of spectral and spatial features. Hu et al. [20] propose a pyramid neural architecture search network to automatically design hyper-parameters such as scales, patches, and standard cell operators in the multi-stage deblurring network. Moreover, Liu et al. [21] propose an image-deblurring framework by using pixel loss combined with a high-frequency loss to train their model to further improve the deblurring performance. This framework comprises a high-frequency reconstruction network and a multi-scale grid network. Despite the recent progress of the above multi-stage methods, they mainly train their models based on pixel loss and still suffer from the issue of over-smoothed outputs.

Recently, GAN-based methods [25,26,27] have also been used in motion deblurring. In particular, Kupyn et al. [25] present DeblurGAN based on conditional motion deblurring. DeblurGAN uses the generator to restore sharp images from synthetic and real-world blurry images and makes the discriminator unable to distinguish the generated image from real, sharp images. Based on DeblurGAN, Kupyn et al. [26] further devise an improved version (i.e., DeblurGAN-v2). They employ a relativistic discriminator that incorporates global and local scale evaluations. Another GAN-based method [27] first uses leaning-to-blur GAN (BGAN) to generate photo-realistic blurry images by learning features from real-world blurry images, then leverages a learning-to-deblur GAN (DBGAN) to recover sharp images from these blurry images. However, they usually require multiple losses and careful tuning of parameters. Moreover, due to the lack of pixel distance constraints, it is easy to cause different degrees of distortion in the output results.

### 2.2. Attention Mechanisms in Image Deblurring

With the recent success of transformer architecture [32] in natural language processing, attention mechanisms have been introduced into image processing tasks [30,33,34]. Recently, there also has been application of attention mechanisms to image deblurring [14,16,17]. In particular, Shen et al. [14] propose a deblurring method consisting of three separate branches for removing foreground humans, background, and global blur. They present a human-aware attention module to focus on the deblurring for the location of people because the image regions of the foreground human are usually the most attentive. Suin et al. [16] propose using both the global attention and adaptive local filters to learn the transformation of features to improve deblurring performance. Moreover, Purohit et al. [17] propose using self-attention to calculate all spatial location correlations for learning non-local connections between features at different locations.

## 3. Methodology

This section presents MSAN architecture, attention mechanisms in MSAN, and the used loss functions for our method.

### 3.1. MSAN Architecture

Figure 1 presents the overall architecture of the proposed MSAN, which consists of four sub-networks with the same structure. These sub-networks progressively process deblurring from top to bottom. The input scale of each sub-network is the same. The first sub-network uses the original blurred image as input, and the second, third, and fourth sub-networks use the concatenation of the original blurred image and the previous sub-network’s output as input. By adding the last sub-network’s output to the original blurred image, the final deblurred image is obtained.

Each sub-network contains an encoder, a decoder, and a self-attention module. In particular, the encoder has three convolutional layers. The first convolutional layer transforms the input into feature maps of 32 channels. The second and third convolutional layers downsample the feature map scale by half, and output feature maps of 64 and 128 channels, respectively. Each convolutional layer is followed by four ResBlocks for feature extraction. A self-attention module is placed between the encoder and decoder to recalibrate the extracted features. A decoder has two deconvolutional layers. These two convolutional layers upsample the feature map scale by two, and output feature maps of 64 and 32 channels, respectively. Each deconvolutional layer is followed by four ResBlocks for feature reconstruction. The last convolution of the decoder transforms the feature maps into a three-channel output.

### 3.2. Attention Mechanisms in MSAN

The main purpose of attention mechanisms is to get the most useful and important features by calculating the correlation of relevant positions. These features are then recalibrated to enhance the adaptability of the model. Inspired by the work of [17,30], we design a SAM that is more suitable for our model based on the self-attention mechanism to improve the adaptability of the model to different blurry image datasets. Because self-attention requires a great deal of memory, the hardware cannot meet the computational demand when the height and width of the input feature map are too large. We introduce group convolution to downsample the input of SAM to reduce memory footprint. Meanwhile, we place SAM between the encoder and decoder to further reduce memory usage. We next present the details of SAM as follows.

As illustrated in Figure 2, the feature map $X\in R^{\hat{C}\times \hat{H}\times \hat{W}}$ is the output from the encoder, where $\hat{C}$, $\hat{H}$, and $\hat{W}$ are the channels, height, and width of the feature map X, respectively. Different from self-attention in [30], we use group convolution to downsample X and get $X^{\prime }\in R^{\hat{C}\times \frac{\hat{H}}{2}\times \frac{\hat{W}}{2}}$. In this way, both the height and width of the feature map are reduced by half to decrease the computing cost of self-attention.

Next, $X^{\prime }$ is transformed into three feature spaces Q, K and V, where $Q,K,V\in R^{\hat{C}\times \frac{\hat{H}\hat{W}}{4}}$. We get $Q^{T}$ by transposing Q, the $i^{th}$ row of $Q^{T}$ represents the values of all channels at the *i* position on $X^{\prime }$. And the $j^{th}$ column of K represents the values of all channels at the $j^{th}$ position on $X^{\prime }$. We calculate the position relationship by performing matrix multiplication between $Q^{T}$ and K. The product result uses the softmax layer to calculate the attention map $\in R^{\frac{\hat{H}\hat{W}}{4}\times \frac{\hat{H}\hat{W}}{4}}$ as follows,
(2)$$a_{ji}=\frac{\exp({\left(Q^{T}K\right)}_{ij})}{\sum_{j=1}^{\frac{\hat{H}\hat{W}}{4}}\exp\left({\left(Q^{T}K\right)}_{ij}\right)},$$
where $a_{ji}$ denotes the extent of correlation between the *i* position and the *j* position. A greater correlation between any two positions means greater similarity (greater attention) between their feature representations.

We transpose A to obtain $A^{T}$. The matrix multiplication between V and $A^{T}$ produces an enhanced feature-map residual $\tilde{O}\in R^{\hat{C}\times \frac{\hat{H}\hat{W}}{4}}$. We then transform the shape of $\tilde{O}$ to get $O\in R^{C\times \frac{\hat{H}}{2}\times \frac{\hat{W}}{2}}$ and use group convolution to upsample O to obtain $Y^{\prime }\in R^{\hat{C}\times \hat{H}\times \hat{W}}$. Finally, we multiply $Y^{\prime }$ by parameter λ to add with X. Thus, the final output denoted by $Y\in R^{\hat{C}\times \hat{H}\times \hat{W}}$ is as follows,
(3)$$Y=\lambda Y^{\prime }+X,$$
where λ is a learnable scalar, which is zero at the initial stage, and the attention module directly returns X. By training, the attention module gradually learns to add the weighted $Y^{\prime }$ to the original X, thus realizing the adaptive modulation of the deblurring features.

The SAM enhances relevant features and reduces irrelevant features based on the selective aggregation of self-attention so that the model can focus on processing more critical features. In addition, different types of features in different images have various contributions to their deblurring. SAM re-calibrated the extracted features (i.e., ignoring irrelevant information in the features and focusing on their important information), making these features more suitable for the decoder. Therefore, the adaptability of the whole model to different blurred images is improved.

### 3.3. Loss Function

Most MSN methods employ pixel loss such as MSELoss to train their models. These models are optimized by MSELoss between ground truth images and predicted results, which are able to produce better results under the evaluation of peak signal-to-noise ratio (PSNR). This is because there has been a negative correlation between metric and loss: (4)$$PSNR=10\times \log_{10}\left(\frac{{\left(2^{8}-1\right)}^{2}}{MSE}\right).$$
Thus, the PSNR value between the deblurring images and the ground-truth images is maximized when the MSELoss is minimized.

However, as mentioned earlier, models optimized directly by using pixel loss are prone to over-smoothed outputs. To reduce the impact of pixel loss while maintaining a good PSNR, we propose a novel strategy based on binary cross-entropy loss to optimize our multi-stage network. In particular, the definition of the binary cross-entropy loss function is as follows,
(5)$$BCELoss=-\frac{1}{N}\sum_{i=1}^{N}\left(y_{i}\log\left(p(y_{i})\right)+\left(1-y_{i}\right)\log\left(1-p(y_{i})\right)\right),$$
where $y_{i}$ is the label (1 for class I and 0 for class II), expressions $p(y_{i})$ and $1-p(y_{i})$ are the predicted probability of class I and class II of the $i^{th}$ sample, and N is the total number of samples. In particular, we denote the output D of MSAN to represent a set including C×H×W samples (where *C*, *H*, and *W* represent the channel number, height, and width of D), and define the two labels, class I and class II as blur (y=1) and sharp (y=0), respectively. Then, we use S to denote the ground-truth image and $r_{i}$ to denote the difference between deblurring image D with ground-truth image S at the $i^{th}$ pixel position, where $r_{i}$ is defined as follows: (6)$$r_{i}=|D_{i}-S_{i}|.$$
Our approach assumes that the predicted probability of blur label of the $i^{th}$ sample is linearly related to $r_{i}$. When $r_{i}$ is larger, the probability that the sample is blur labeled is greater. The absolute difference between two pixels (i.e., $r_{i}$) is at most 255 due to the value of each pixel ranging from 0 to 255. We thus denote the probability that the $i^{th}$ sample is a blur label as follows,
(7)$$p\left(y_{i}\right)=\frac{r_{i}}{255}=\frac{\left|D_{i}-S_{i}\right|}{255}.$$
Therefore, Equation (Equation 5) used for our model can be rewritten as follows: (8)$$BCELoss=-\frac{1}{C\times H\times W}\sum_{i=1}^{C\times H\times W}\left(y_{i}\log\left(\frac{\left|D_{i}-S_{i}\right|}{255}\right)+\left(1-y_{i}\right)\log\left(1-\frac{\left|D_{i}-S_{i}\right|}{255}\right)\right).$$

Because the sample set output by MSAN is still the set of pixel values of the deblurring image D, the corresponding image is sharper when the probability of the pixel values being predicted as sharp labels is greater. In particular, we predict each sample from D as a sharp label without considering the case of a blur label, i.e., $y_{i}$ is always equal to 0. Therefore, the loss function of MSAN can be defined as follows: (9)$$L_{MSAN}\left(D,S\right)=-\frac{1}{C\times H\times W}\sum_{i=1}^{C\times H\times W}\log\left(1-\frac{\left|D_{i}-S_{i}\right|}{255}\right).$$

The optimization goal of our model is to reduce the loss (i.e., $L_{MSAN}\left(D,S\right)$). As shown in Figure 3, the smaller value of the expression $\log\left(1-\frac{\left|D_{i}-S_{i}\right|}{255}\right)$, the closer $D_{i}$ is to $S_{i}$, indicating the effect of the deblurring image is better.

## 4. Experiments

In this section, we demonstrate the effectiveness of the proposed MSAN. We first present the experimental implementation. We then provide a detailed performance comparison between our method and other SOTA methods in both quantitative evaluations and visual comparisons. Finally, we investigate the contributions of SAM and loss function in the proposed MSAN by ablation study.

### 4.1. Implementation

#### 4.1.1. Datasets

We used three representative datasets (i.e., GoPro [10], HIDE [14], and RealBlur [35]) for experiments. The GoPro and HIDE datasets use high-frame-rate cameras to capture high-frame-rate image sequences, which are then stacked to synthesize both blurred and sharp images aligned at pixel level. The RealBlur dataset captures blurred and sharp image pairs simultaneously by using a dual-camera system consisting of a beam splitter and two cameras. Whereas HIDE is divided into HIDE I for objects with long-shot depth and HIDE II for objects with close-up depth, and ReadBlur is divided into RealBlur-R from raw images and RealBlur-J from JPEG images. We follow the same configuration as [10] and all models are trained with 2103 image pairs from the GoPro dataset [10]. For generalization testing, we use several testing datasets: 1111 image pairs in the GoPro dataset [10], 1063 image pairs in the HIDE I dataset [14], 962 image pairs in the HIDE II dataset [14], 980 image pairs in the RealBlur-R dataset [35] and 980 image pairs in the RealBlur-J dataset [35].

#### 4.1.2. Training Details

Our model is implemented by using the PyTorch [36] library. The source code and model are available at: https://github.com/CaiGuoHS/MSAN, accessed on 21 September 2022. For fairness, unless noted otherwise, all comparative experiments are based on the Cuda toolkit 10.2 and conducted on the same machine with a single NVIDIA RTX 2080Ti GPU. We use the Adam solver to optimize the MSAN, and train the model on 3000 epochs with 256×256 pixel images and a batch size of 8. In the initial warming-up phase [37], the initial learning rate of MSAN is set to $10^{-4}$, then gradually reduced to $10^{-6}$ by using the cosine annealing procedure. Furthermore, we use the same techniques to process the training data as we do for data augmentation.

### 4.2. Comparison with SOTA Methods

We compare our method with other SOTA motion-deblurring methods, including SRNDeblur [11], DeblurGAN-v2 [26], DSDeblur [12], Stack(4)-DMPHN [13], MTRNN [15], and HFRSN-MSGSN [21]. For comparison fairness, the experimental results from other comparative approaches are obtained by the source code and pre-trained models of these methods.

#### Quantitative Evaluations

We utilize PSNR and structural similarity index measure (SSIM) as metrics for quantitative evaluations. In general, PSNR and SSIM scores that are higher indicate better deblurring. We first quantitatively evaluate the comparative methods on the GoPro [10] testing set. In addition to PSNR and SSIM, we also compare the model size and average processing time per image for each method, as shown in Table 1. We observe that the proposed MSAN reaches the highest scores, 32.24 and 0.956, in terms of PSNR and SSIM, respectively, and its model size and running time rank second and third among these methods. The second-best model for PSNR score is HFRSN-MSGSN [21], which has 0.39 dB less PSNR than ours, and its model size and runtime are three times and twice that of our MSAN, respectively. Likewise, the third-best one for PSNR score (i.e., Stack(4)-DMPHN [13]) has a much larger model size and runtime than our MSAN. Although DeblurGAN-v2 achieves a running time of 142 ms (i.e., the best), it uses Inception-ResNet-v2 [38] as the backbone, making its model size much larger than other methods. Although MTRNN has the smallest model size, its recurrent network architecture makes its running time still larger than our MSAN. In contrast, our MSAN has obvious advantages over other SOTA models because its running time is slightly higher than 400 ms, and its model size is less than 36 MB. The methods in [11,12] use multi-scale inputs to increase the receptive field to restore blurry images and spend more computing time than DeblurGAN-v2 [26].

Table 2 shows PSNR and SSIM values on HIDE and RealBlur testing sets for the generalization tests. The best results are highlighted in bold. We observe that the ranking of PSNR and SSIM scores on the HIDE and RealBlur testing sets by other methods differs from the ranking on the GoPro testing set. The second-best model (i.e., HFRSN-MSGSN [21]) on the GoPro is also ranked second on the HIDE (including HIDE I and HIDE II) but performs poorly on the RealBlur, especially on the RealBlur-J. The third-best one (i.e., Stack(4)-DMPHN [13]) on the GoPro is the third-ranked only on the HIDE II, and the performance on other datasets is mediocre, even the last one on the RealBlur-J. On the contrary, despite DeblurGAN-v2 being the last one on the GoPro, HIDE I, HIDE II, and RealBlur-R, it ranks second on the RealBlur-J. In contrast, our model performs the best on all testing sets, indicating that our model generalizes better than other models on these datasets.

### 4.3. Visual Comparisons

We visually compare the deblurring results of each method on different testing sets. Because the blurred input image has artifacts and most of the methods trained by MSE loss tend to smooth the input artifacts while performing deblurring, the deblurring results obtained by these methods are still not sharp enough. In contrast, our method trained by BCE loss reduces these over-smoothing effects in the output, making the deblurring results sharper.

We choose three images from the GoPro testing set and divide the deblurring results into three columns. As shown in Figure 4, although all methods achieve deblurring, our MSAN model reduces over-smoothing effects in output and produces the best image quality compared to other methods. In particular, the letters and license plate numbers of the magnified parts in the first and second columns, the deblurring results of our method have fewer artifacts and sharper images. Moreover, for the windows in the magnified part of the third column, we can also observe that our method handles the details better than other methods. For example, our deblurring result is more accurate for the restoration of the edge of the window and the shadow part on the right.

We also compare the deblurring results of our MSAN and these SOTA methods on HIDE I, HIDE II, RealBlur-R, and RealBlur-J testing sets, as shown in Figure 5. Because our method generalizes better than other methods, we can observe that our MSAN model achieves significantly better deblurring results than other models on the generalization test. Compared to other deblurring methods, our deblurring results are affected by relatively little over-smoothing, so they are much sharper and closer to ground-truth images. For example, in the magnified parts in the first and third columns, only our MSAN restores the correct letters, especially the tiny letters in the third column. Likewise, for the zoomed-in sections in the second and fourth columns, only our MSAN restores accurate detail.

### 4.4. Ablation Study

We also carry out ablation experiments to evaluate the effectiveness of the proposed SAM and the BCELoss strategy. We examine the following three variants of MSAN:M1 (MSAN without SAM and uses MSELoss);M2 (MSAN without SAM and uses BCELoss);M3 (MSAN uses MSELoss).

We retrained models M1, M2, and M3, respectively. The quantitative evaluation of ablation experiments is shown in Table 3. We can observe that in the generalization test (i.e., based on HIDE I, HIDE II, RealBlur-R, and RealBlur-J testing sets), the model with SAM outperforms the model without SAM (i.e., M3 > M1 and MSAN > M2). This shows that the proposed SAM can effectively improve the generalization ability of the model. For different loss strategies, in the case without SAM (i.e., M1 and M2), M2 using BCELoss slightly outperforms M1 using MSELoss on most testing sets. When SAM is used (i.e., M3 and MSAN), MSAN using BCELoss is significantly better than M3 using MSELoss. This shows that the proposed BCELoss strategy can effectively improve the performance of the model, especially for the model using SAM. We further compare the two loss strategies. As shown in Figure 6, we quantitatively evaluate two strategies on different testing sets every 200 epochs, and we can see that MSAN using BCELoss offers better performance.

## 5. Conclusions

We present a motion deblurring model, MSAN, that is efficient and has good generalization performance for handling motion-blur images. We introduce a new self-attention module to the encoder–decoder to effectively reduce the computing cost and improve the model’s adaptability to different blurred images. Furthermore, the proposed binary cross-entropy loss strategy is more suitable for training the deblurring model than the pixel-loss strategy and effectively improves the deblurring performance of the model. Extensive experiments on several benchmark datasets demonstrate that MSAN achieves SOTA performance for motion deblurring tasks.

## Figures and Tables

**Figure 1 entropy-24-01414-f001:**
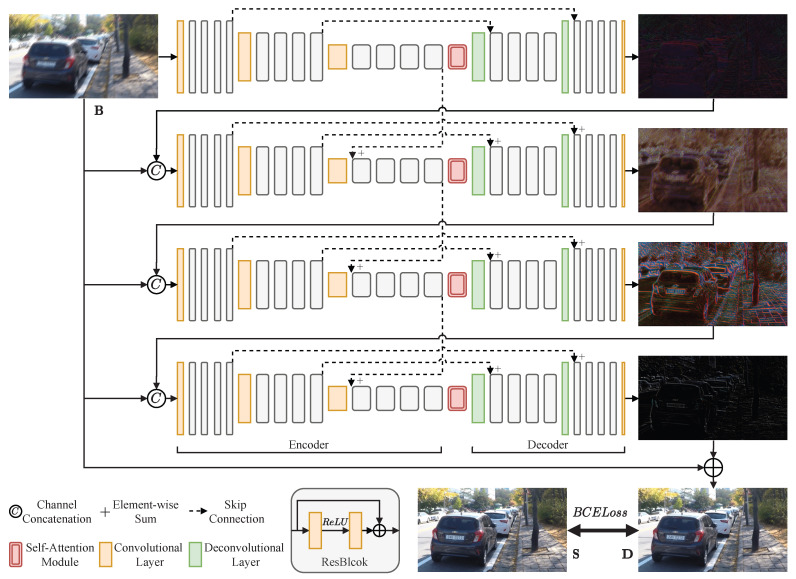
Illustration of MSAN architecture. Our MSAN contains four sub-networks; these sub-networks process deblurring from top to bottom progressively. The terms B, D, and S denote the original blurry image, the deblurred image, and the sharp image, respectively.

**Figure 2 entropy-24-01414-f002:**
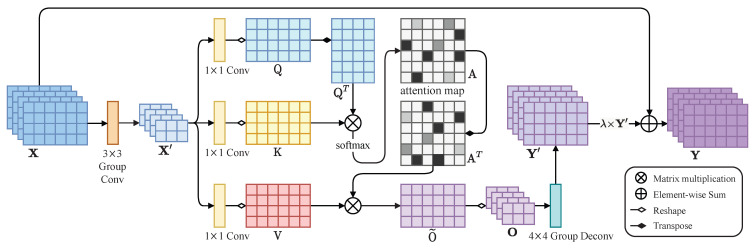
Illustration of the proposed SAM. $X^{\prime }$ is obtained by downsampling (using group convolution) the feature map *X* and transforming $X^{\prime }$ into three feature spaces Q, K, and V. Next, we calculate the similarity between Q and K to obtain the weights and use the softmax function (performed on each row) to normalize these weights. We then perform a weighted summation of the weights and the corresponding V to obtain O and use group convolution to upsample O to obtain $Y^{\prime }$. Finally, we multiply $Y^{\prime }$ by parameter λ to add with X to get the final output Y.

**Figure 3 entropy-24-01414-f003:**
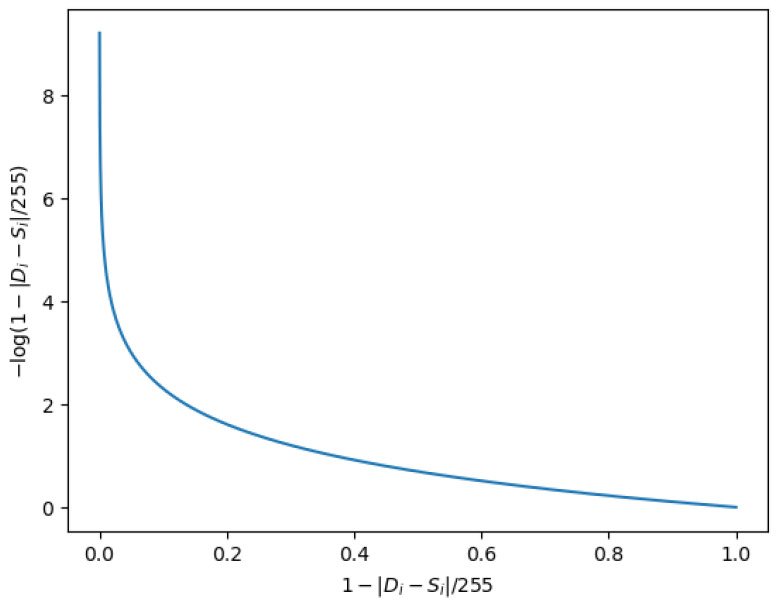
Loss function curve.

**Figure 4 entropy-24-01414-f004:**
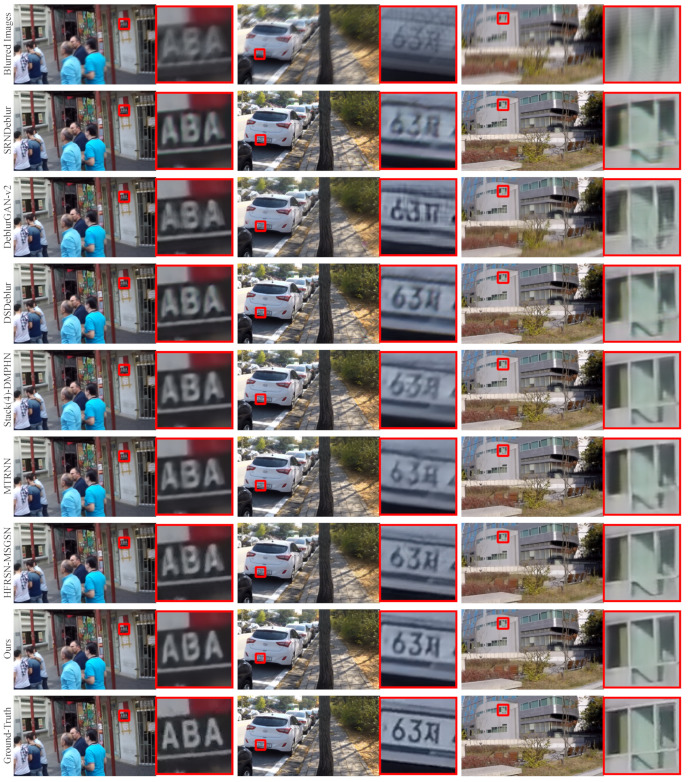
Visual comparison of deblurring results of MSAN and other methods on the GoPro dataset. All three images are from the GoPro dataset. The first and last rows show blurred images and ground-truth images, respectively. The second to seventh rows from top to bottom are deblurring images obtained by SRNDeblur [11], DeblurGAN-v2 [26], DSDeblur [12], Stack(4)-DMPHN [13], MTRNN [15], HFRSN-MSGSN [21], and our MSAN, respectively. A partially magnified image also accompanies each image (see red box).

**Figure 5 entropy-24-01414-f005:**
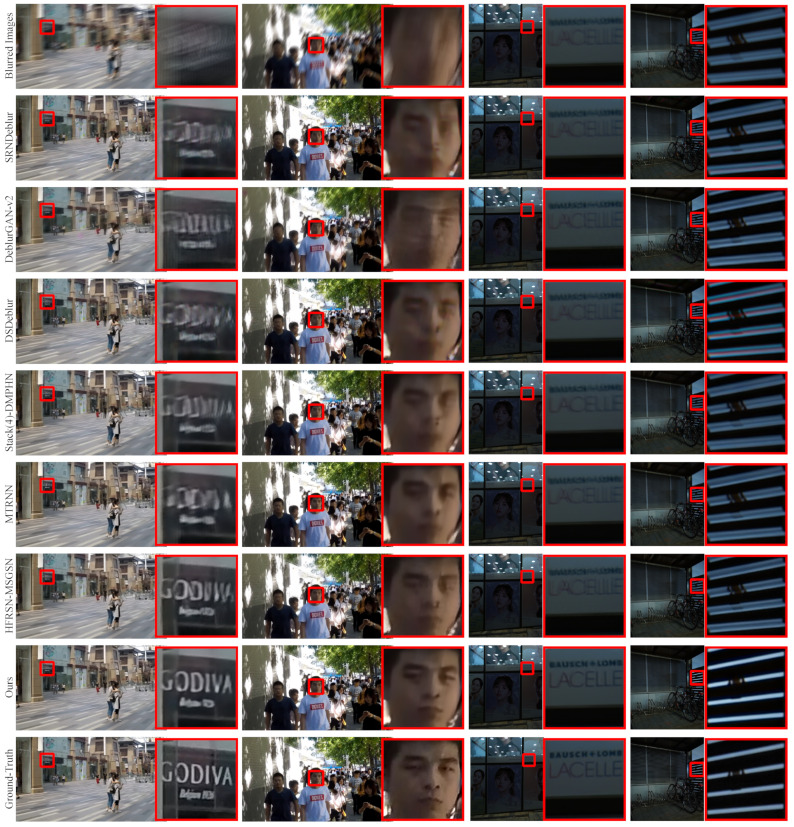
Visual comparison of deblurring results of MSAN and other methods on the HIDE and RealBlur datasets. The four images from left to right are from HIDE I, HIDE II, RealBlur-R, and RealBlur-J datasets. The first and last rows show blurred images and ground-truth images, respectively. The second to seventh rows from top to bottom are deblurring images obtained bySRNDeblur [11], DeblurGAN-v2 [26], DSDeblur [12], Stack(4)-DMPHN [13], MTRNN [15], HFRSN-MSGSN [21], and our MSAN, respectively. A partially magnified image also accompanies each image (see red box).

**Figure 6 entropy-24-01414-f006:**
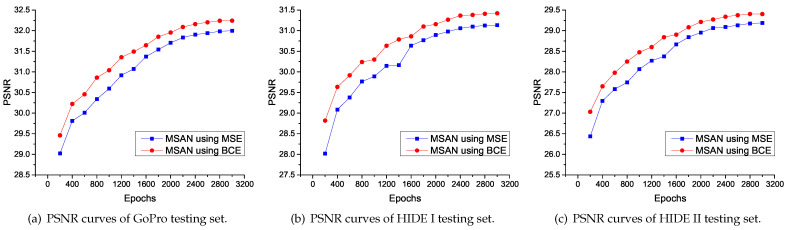
PSNR curves of various tests of MSAN using different loss strategies.

**Table 1 entropy-24-01414-t001:** Quantitative evaluation of MSAN and other methods using the GoPro testing set. The best scores are highlighted in **bold**.

Models	GoPro			
	PSNR	SSIM	Model Size	Time ^1^
SRNDeblur [11]	30.20	0.933	33.6 MB	814 ms
DeblurGAN-v2 [26]	29.08	0.918	244.5 MB	**142 ms**
DSDeblur [12]	30.96	0.942	49.8 MB	805 ms
Stack(4)-DMPHN [13]	31.39	0.948	86.9 MB	637 ms
MTRNN [15]	31.13	0.945	**10.6 MB**	492 ms
HFRSN-MSGSN [21]	31.85	0.951	108.3 MB	899 ms
VDN [23]	31.65	0.951	23.4 MB	236 ms
MSAN (Ours)	**32.24**	**0.956**	35.9 MB	419 ms

^1^ The time is an average time taken to deblur images on the GoPro testing set. To measure this time we use torch.cuda.synchronize() because PyTorch’s CUDA calls are asynchronous.

**Table 2 entropy-24-01414-t002:** Quantitative evaluation of MSAN and other methods using the HIDE and RealBlur testing sets. The best scores are highlighted in **bold**.

Models	HIDE I		HIDE II		RealBlur-R		RealBlur-J	
	PSNR	SSIM	PSNR	SSIM	PSNR	SSIM	PSNR	SSIM
SRNDeblur [11]	29.18	0.912	27.46	0.891	35.27	0.937	28.48	0.861
DeblurGAN-v2 [26]	28.29	0.896	26.65	0.872	35.11	0.935	28.69	0.866
DSDeblur [12]	29.98	0.923	28.14	0.902	35.39	0.938	28.39	0.859
Stack(4)-DMPHN [13]	29.80	0.925	28.34	0.910	35.48	0.947	27.80	0.847
MTRNN [15]	29.98	0.926	28.24	0.908	35.79	0.951	28.44	0.862
HFRSN-MSGSN [21]	30.30	0.929	28.58	0.911	35.40	0.934	28.31	0.854
VDN [23]	30.09	0.929	28.43	0.912	35.54	0.948	28.18	0.855
MSAN (Ours)	**31.42**	**0.944**	**29.40**	**0.926**	**35.96**	**0.953**	**28.83**	**0.878**

**Table 3 entropy-24-01414-t003:** Ablation experiments for MSAN on the GoPro testing set. The best scores are highlighted in **bold**.

Models	MSE	BCE	SAM	GoPro		HIDE I		HIDE II		RealBlur-R		RealBlur-J	
				PSNR	SSIM	PSNR	SSIM	PSNR	SSIM	PSNR	SSIM	PSNR	SSIM
M1	✓			32.06	0.953	30.40	0.932	28.81	0.916	35.55	0.947	28.25	0.857
M2		✓		32.18	0.955	30.69	0.936	28.92	0.919	35.53	0.946	28.30	0.860
M3	✓		✓	32.00	0.953	31.13	0.940	29.19	0.921	35.91	0.953	28.74	0.874
MSAN		✓	✓	**32.24**	**0.956**	**31.42**	**0.944**	**29.40 **	**0.926**	**35.96**	**0.953**	**28.83**	**0.878**

## Data Availability

Not applicable.

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
