# Peer review of "Multi-Stage Attentive Network for Motion Deblurring via Binary Cross-Entropy Loss"

_entropy, 2022, doi:10.3390/e24101414_

Round 1

Reviewer 1 Report

The revised manuscript proposes a new learning-based method for image deblurring employing multiple encoder-decoder sub-models with self-attention layers and fitted with a binary cross-entropy loss function. The introduction correctly mentions already reported state-of-the-art methods for image deblurring techniques. The authors state the actual contribution of the proposed model. The proposal is clearly described, and its performance is properly contrasted against other state-of-the-art models. Overall, the reported results are slightly better than other tested methods, with an apparent better generalization (especially for the HIDE and RealBlur datasets). A final ablation study confirms that the proposed model using a binary cross-entropy loss function offers better performance than the model fitted with a pixel loss function. Although the manuscript is suitable for publication in this Journal, I still have some comments on the revised manuscript that must be addressed before publication:

1.      Although the reason to use a binary cross-entropy loss function relies on the fact that models using pixel loss functions "are prone to over-smoothed outputs", output images in these conditions are not fully described and presented in the manuscript. A further discussion is required involving over-smoothed output images by other models and how the proposal performs over these same images.

2.      The authors validated their proposal against other methods; however, did the authors compare their proposal against their recently published paper: https://onlinelibrary.wiley.com/doi/10.1002/cav.2066

What is the difference when compared to this proposal by the same authors? These issues should be addressed before publication.

3.      In the introduction and the conclusions, the authors claim that the inserted self-attention module "effectively reduces the computing cost". It is not clear what the authors mean by this statement. The proposal does not provide the fastest training times when observing the results in Table 1. Further discussion on this issue is required.

4.      I understand that the authors have tested their proposal using isolated images extracted from videos. Nevertheless, deblurring a whole video sequence will provide enough experimental validation of the claimed "good generalization performance". The latter, if we consider that not only spatially-varying features within each image but also time-varying features within actual image sequences in videos, must be solved to guarantee proper "motion deblurring".

5.      The authors provided the source code, which I believe is a good practice. However, I could not check its functioning because at running the test script, an error popped up:

FileNotFoundError: [Errno 2] No such file or directory: './ckpts/MSAN.pth'

I did not find that file in the GITHUB repository. The authors must either correct this error before publication or further explain the technical details of their implementation.

 Minor comments:

- Check the ORCID link for the last author.

- For completeness, some references are missing in some paragraphs/sentences:

* "Motion blur removal is a challenging problem in vision tasks due to motion blurs are spatially varying blurs caused by various factors, e.g., camera shaking or objects quickly moving."

* "particular, the most essential reason for motion blur is that the large-distance relative motion of objects in the shooting scene causes the originally collected pixels to change during the exposure time, and the variation of these pixels is complex and non-uniform."

*"A later study by Gao et al. suggested that a parameter selective sharing scheme in combination with a nested skipping structure o...."

Reviewer 2 Report

Despite the authors claim, there are few publications using CNN for motion deblurring using Binary Cross Entropy Loss.

Authors have not referred to adequate number of references on CNN based deblurring methods

Structural Similarity Index Measure (SSIM) is a poor choice as blurring does not change the structure of an image. If you estimate SSIM and then blur the image and then estimate the SSIM, the values will not differ by much as the structure remains intact.

Round 2

Reviewer 2 Report

I believe the authors have addressed most of my concerns.